# Pretrained Vision Models for Predicting High-Risk Breast Cancer Stage

**Bonaventure F. P. Dossou** [1234]**, Yenoukoume S. K. Gbenou**[5]**, Miglanche Ghomsi Nono**[6]

[1]Center for Intelligent Machines, McGill University, [2]Mila Quebec AI Institute, [3]Lelapa AI,
[4]Masakhane Research Foundation, [5]Drexel University, [5]University Of Maryland Baltimore County

## Abstract

Cancer is increasingly a global health issue. Seconding cardiovascular diseases, cancers are the second biggest cause of death in the world with millions of people succumbing to the disease every year. According to the World Health Organization (WHO) report, by the end of 2020, more than 7.8 million women have been diagnosed with breast cancer, making it the world's most prevalent cancer. In this paper, using the Nightingale Open Science dataset of digital pathology (breast biopsy) images, we leverage the capabilities of pre-trained computer vision models for the breast cancer stage prediction task. While individual models achieve decent performances and demonstrate usefulness to the task at hand, we find out that the predictions of an ensemble model are more efficient.

## 1 Introduction

The treatment of Breast Cancer (BC) can be efficient when the disease is detected at a very early stage, and there are mainly five stages of BC (Stage 0, Stage 1, Stage 2, Stage 3, and Stage 4). One of the most popular ways of detecting BC is breast biopsy; a process during which, samples of cell tissues are collected to be examined in the laboratory with a microscope, in order to locate the presence, cause, or extent of the disease. Thanks to recent advances in Artificial Intelligence (AI), especially Deep Learning (DL), there has been a rise in research efforts to leverage the potential of DL-based systems to help in breast cancer detection. These recent initiatives made use of pretrained Computer Vision (CV) models and were applied to Full-Field Digital Mammography (FFDM) in which X-rays mammograms are converted into electrical signals, for a binary classification (detecting BC or not BC) setting. In this work, after briefly describing the Nightingale Open Science Dataset of Digital Pathology (NOSDDP) Bifulco et al. (2021); Mullainathan & Obermeyer (2022), we use slides from each biopsy digital image and pretrained CV models, to predict the stage of cancer of a patient. We also explore the efficiency of deep ensembles.

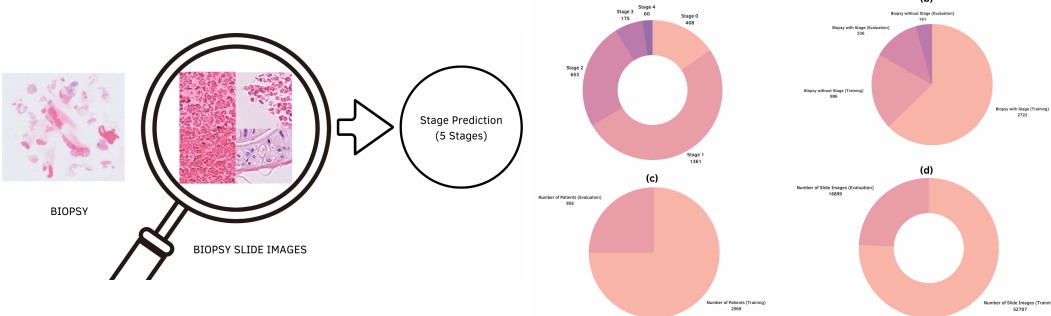

Figure 1: (left) Overview of the task. (right) Statistics of the dataset: (a) biopsies per cancer stage, (b) labeled biopsies and non-labeled biopsies across the different dataset splits, (c) patients used for each dataset split, and (d) slide images across splits.

## 2 DATASET, EXPERIMENTS AND RESULTS

Existing works have demonstrated the ability of DL-based systems to predict the type of cancer. However, there are currently very few to no datasets, that link biopsy images to the cancer stage of the patients, and the respective outcomes. As an attempt to bridge that gap, the NOSDDP contains 72400 biopsy slide images, from 4335 breast biopsies of 3425 unique patients, observed from 2014-2020. These slide images are linked to cancer stages and information about the mortality of patients (Figure 1 (left)). Figure 1 (right) presents the statics across the dataset, and its different splits (train and evaluation). We can see that Stage 1 is the most frequent cancer stage across labeled biopsies. It is also important to notice that there are biopsies without stages that we further labeled as Stage 1, to make use of all available data.

We downsampled the high-resolution slide images to (224, 224), as supported by many existing pretrained CV models. The original training set has been split into two subsets (with 80:20 ratio): $D_{Train}$ and $D_{Eval}$, while the initial evaluation set is used as a test dataset, we denote it $D_{Test}$. Image samples from $D_{Train}$ have been augmented using randomized cropping and horizontal flipping, then normalized. The image samples from $D_{Eval}$ and $D_{Test}$ have only been normalized. We finetuned 10 pretrained CV models: Resnet (18, 50, 152), EfficientNet Tan & Le (2021), ConvNext Liu et al. (2022) (M), WideResNet Zagoruyko & Komodakis (2016) (101), VGG Simonyan & Zisserman (2014), ResNext Xie et al. (2017) (101), RegNet Radosavovic et al. (2020) (X32GF), Swin Transformer Liu et al. (2021) (B), and MaxVit Tu et al. (2022). For each model, we try various learning rates, with the *AdamW* optimizer. Each model was trained with a batch size of 32, and for 50 epochs. The predicted cancer stage $PCS$ for a biopsy image $B$ is the average of the predicted cancer stage for each subsequent slide image $S$: $PCS(B) = \frac{1}{|B|} \sum_i PCS(S_i)$, where $|B|$ is the number of slides for the biopsy $B$, and $PCS(S_i) \in \{0, 1, 2, 3, 4\}$: this implies that the prediction for a biopsy image is a continuous value $\in [0, 4]$. Therefore, to measure the closeness to the true label which $\in \{0, 1, 2, 3, 4\}$, we use the Mean Square Error (MSE) metric. The different performances are summarized in Table 1.

| Learning Rate | Resnet 18 | Resnet 50 | Resnet 152 | EfficientNet (M) | ConvNext (Base) | Wide Resnet 101 | VGG | RegNet | SwinT (B) | MaxVit |
|---|---|---|---|---|---|---|---|---|---|---|
| 1e-4 | 1.009611 | **0.970504** | 1.005862 | 0.925831 | 1.009943 | **0.931212** | **0.939045** | 0.992109 | 0.898370 | **0.855656** |
| 1e-5 | **1.001620** | 1.008508 | **1.001902** | 0.987911 | **0.957443** | 1.012101 | 0.994687 | **0.988756** | **0.897878** | 0.893357 |
| 4e-4 | 1.020936 | 0.986704 | 1.007281 | **0.829745** | 1.110100 | 1.033937 | 1.107765 | 1.000450 | 1.231371 | 1.098370 |

Table 1: MSE of each pretrained CV model. **The bold numbers represent the best performance of each model, with respect to learning rates**.

EfficientNet performs better, due to its architecture, which combines several neural networks (like a deep ensemble (DE)), optimizing their respective accuracy and efficiency via neural architecture search Tan & Le (2021)[1]. Following the intuition of EfficientNet, we also tried to create a DE ($E$) of individual pretrained CV models. The predicted cancer stage of the ensemble model $E$, for a biopsy $B$ is $PCS(B)_E = \frac{1}{|E|} \sum_{k=1}^{|E|} PCS(B)_k$, where $PCS(B)_k$ is the $PCS(B)$ of the $k$-th model, and $|E|$ is the size of $E$ i.e. the number of models composing $E$.

We explore two strategies: (a) $E$ with all models, and (b) $E$ solely the models $M$ such that $MSE(M) < 1$. Setup (b) with **0.5543481** performs better than (a) with MSE *0.632767*, showing that DEs are better than individual models. This makes sense because in real-life, aggregating the predictions of many doctors is better than relying on a prediction from a single doctor. Moreover, each model learns different representations and features, which enriches the representation of $E$. We also could notice that the lower the MSE of individual models, the lower the MSE of $E$.

## 3 CONCLUSION

In this work, we explored the applicability of pretrained computer vision models, in the task of predicting high-risk breast cancer. Our experiments demonstrated that pretrained models are indeed useful for the task at hand. Moreover, we found that deep ensemble models offer more significant performances, compared to single models.

---

[1]EfficientNet: Improving Accuracy and Efficiency through AutoML and Model Scaling (GoogleAI Blog): https://ai.googleblog.com/2019/05/efficientnet-improving-accuracy-and.html

## URM STATEMENT

The authors of this work acknowledge that they all meet the URM criteria of the ICLR 2023 Tiny Papers Track.

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
