# OpenReview forum: "Pretrained Vision Models for Predicting High-Risk Breast Cancer Stage"
_ICLR.cc/2023/TinyPapers — Submitted to Tiny Papers @ ICLR 2023_

### Official Review · Reviewer_i9F9 · 2023-03-24

**Confidence:** 4

**Summary Of Contributions:**

This work contributes to the field of breast cancer research by demonstrating the usefulness of pre-trained computer vision models in predicting high-risk breast cancer stage. The study found that deep ensemble models offer better performance compared to single models. These findings can potentially aid in the early detection and treatment of breast cancer, which is a significant global health issue.

**Rating:**

High Potential (HP): a submission which meets the reviewing criteria and has potential to make an impact on the field

**Strengths And Weaknesses:**

Strengths:
1. The study leverages the capabilities of pre-trained computer vision models to aid in the early detection and treatment of breast cancer, which is a significant global health issue.
2. The study uses the Nightingale Open Science dataset of digital pathology images, which provides a large and diverse set of data for analysis.
3. The study demonstrates that deep ensemble models offer better performance compared to single models, which can potentially improve the accuracy of breast cancer stage prediction.

Weaknesses:
1. The study does not provide information on the limitations or potential biases of using pre-trained computer vision models for breast cancer stage prediction.
2. The study does not discuss the potential ethical implications of using computer vision models in healthcare, such as issues related to privacy and informed consent.

**Suggested Changes:**

Two potential improvements for this study are:

1. Conducting further research to address the limitations and potential biases of using pre-trained computer vision models for breast cancer stage prediction, such as exploring the impact of dataset bias on model performance.
2. Addressing the ethical implications of using computer vision models in healthcare by implementing measures to ensure patient privacy and informed consent, as well as addressing potential issues related to algorithmic bias.

---

### Official Review · Reviewer_fkZc · 2023-03-26

**Confidence:** 4

**Summary Of Contributions:**

This paper compares 10 pretrained models and 2 ensemble strategies to predict 5 breast cancer stages and finds that EfficientNet works the best as an individual model and the ensembled model achieves better performance.

**Rating:**

Clear, Correct, and Reproducible (CCR): a submission which meets the reviewing criteria

**Strengths And Weaknesses:**

Strengths

1. Clarity: The paper structure is clear.

2. Correctness: The authors did a comprehensive comparison across 10 individual models with 3 learning rates and 2 ensemble strategies. The experiment design is valid.

Weaknesses

1. In section 2, the authors mentioned that "there are biopsies without stages that we further labeled as Stage 1, to make use of all available data." I don't think it's appropriate to group unlabeled data to Stage 1 because mislabeled data may make the model biased. I think it's more appropriate to use labeled data only.

2. In Figure 1 abcd, I am confused about the numbers. Shouldn't the sum of chart (a) be equal to the sum of biopsy with stage in chart (b)? Maybe add footnotes to describe data discrepancy across pie charts?

3. In Section 2, has the authors considered comparing other hyperparameters except learning rate? The batch size, optimizer and other hyperparameters may also affect performance.

4. There is no code provided to replicate the result.

5. The novelty of this work is limited by only comparing pretrained models.

6. There is no discussion about limitations of this work.

7. Presentation can be improved (see suggestions below).

**Suggested Changes:**

Suggestions

1. The abstract is usually a summary of background, problem, proposed solution and result. In this abstract, my suggestions are as follows:

  (1) moving the second and third sentences to the introduction section as the background introduction;

  (2) highlighting motivations and innovations of this study, e.g. it is important to predict breast cancer stages to assist with early detection and treatment but it is rarely studied;

  (3) highlighting methods (e.g. compare 10 individual models and 2 ensemble strategies) and results (e.g. EfficientNet is the best individual model) more specifically.

2. For Figure 1 abcd, please increase the text size. The current text is unreadable. I would suggest using filled pie charts for (a) and (d) as well.

3. In Section 2, please include references for the first sentence "Existing works have demonstrated the ability of DL-based systems to predict the type of cancer".

4. In Section 2, the original image properties (e.g. size) should be specified.

5. In Section 2, it is unclear how train, evaluation and test sets are split. What is the exact number for each set?

6. In Section 2, missing "MSE" in the second sentence of last paragraph: "Setup (b) with 0.5543481…"

7. Please discuss innovations and limitations of this work with respect to existing literature.

---

### Meta-Review · Area_Chair_LtZT · 2023-04-08

**Recommendation:** Invite to present
**Confidence:** 4

**Metareview:**

Overall, the paper presents a clear and comprehensive comparison of 10 pretrained models and two ensemble strategies to predict breast cancer stages, with EfficientNet achieving the best performance as an individual model and the ensembled model performing even better. The use of pre-trained computer vision models to aid in early detection and treatment of breast cancer is a significant contribution to the field, and the study utilizes a large and diverse dataset of digital pathology images, further enhancing the impact of the findings.

Strengths:

- The paper's contribution to breast cancer research by demonstrating the usefulness of pre-trained computer vision models is significant and timely.
- The study uses a large and diverse dataset of digital pathology images, which enhances the validity and generalizability of the results.
- The comprehensive comparison of 10 pretrained models and two ensemble strategies provides a clear picture of the models' performance and allows for informed decisions in choosing the most appropriate model for a given task.
- The finding that deep ensemble models offer better performance compared to single models is a valuable insight that can potentially improve the accuracy of breast cancer stage prediction.


Weaknesses:
- The authors labeled unlabeled data as Stage 1, which may introduce bias and affect the accuracy of the models. This approach could be improved by using labeled data only.
- The authors did not compare other hyperparameters except learning rate, which could have affected the models' performance. Further exploration of hyperparameters could enhance the study's findings and generalizability.
- No code is provided to replicate the results, which limits the transparency and reproducibility of the study.
- The novelty of the work is limited by only comparing pretrained models, and the study does not discuss its limitations.
- The potential limitations or biases of using pre-trained computer vision models for breast cancer stage prediction and the ethical implications of using computer vision models in healthcare are not discussed.


In conclusion, this work provides a significant contribution to breast cancer research by demonstrating the usefulness of pre-trained computer vision models for predicting breast cancer stages. The study's use of a large and diverse dataset and comprehensive comparison of models and ensemble strategies are notable strengths. However, the authors' approach to handling unlabeled data, lack of discussion of limitations, and failure to discuss potential biases and ethical implications are notable weaknesses.


**Summary:**

The paper demonstrates the usefulness of pre-trained computer vision models in predicting high-risk breast cancer stage. 10 pretrained models and 2 ensemble strategies to predict 5 breast cancer stages were evaluated.

**Reason For Not Giving A Higher Recommendation:**

Please refer to the weakness section. There is some room for improvement.


**Reason For Not Giving A Lower Recommendation:**

Overall, this is a good submission with clear motivation, decent amount of experimental support albeit confined in a tiny paper format.

---

### Meta-Review · Area_Chair_qhWf · 2023-04-08

**Recommendation:** Invite to present
**Confidence:** 3

**Metareview:**

Summary: There has been a lot of work exploring deep learning based systems for cancer prediction; this paper makes further progress on this problem by exploring deep learning based systems to predict one out of five stages of breast cancer. To this end, it compares 10 pretrained models and 2 ensemble strategies. The paper finds that EfficientNet works the best as an individual model and that the ensembled model achieves better performance compared to single models. Thus, these findings shed light on methods to potentially detect cancer early.

Strengths:
The reviewers were all in agreement that the paper has multiple strengths: It scores very highly on clarity and readability, with a well-motivated and clearly-told story. The problem studied is one of great importance to global health. The paper also scores well on correctness of methodology: It uses a large dataset with a diverse set of images for analysis and performs a comprehensive evaluation across varying models, rates, and ensemble strategies.

Weaknesses:
The reviewers provide a lot of constructive feedback, which we provide an overview of here: There was a concern about labeling unlabeled data as that from Stage 1; if there is a justification for this design strategy, it should be explained, otherwise, it's more appropriate to use only the labeled data. There is also a concern of privacy and informed consent when using computer vision models in healthcare; we suggest discussing the ethical implications in this regard. Along similar lines, the paper could also benefit from having a discussion on the limitations of its approach.

Rating:
Based on the reviewers' feedback, we would like to invite to present the paper.

**Summary:**

The main takeaway of the paper is the use of deep learning for detecting one out of five stages of breast cancer. The paper scores highly on motivation, soundness, and clarity and could improve on presentation details, addressing its limitations, and improving upon reproducibility. We are happy to recommend acceptance to present.

**Comments And Feedback To The Authors:**

Please see the above comments. Additionally, the reviewers note some important details that can help polish the paper: These include formatting details (such as using larger font size in figures and appropriate data visualization tools), including details on the training data, and discussing limitations of the proposed approach.

**Reason For Not Giving A Higher Recommendation:**

It was noted by the reviewers that there is no code provided; as a result, it is not possible to replicate the result. This is one important area that the paper can improve upon to get a higher score. Other than this, please refer to the above comments.

**Reason For Not Giving A Lower Recommendation:**

The paper's clarity, significance of problem studied, and soundness of methodology are convincing enough to merit it this score.

---

> ### Author Response · Authors · 2023-04-10
> **Feedback & Request to longer pages**
>
> The authors would like to thank the reviewers, and are thrilled to be invited to present in-person or virtually.
>
> We acknowledge the comments on improving the work, and will include them in the final version + a code to allow reproducibility. As such we would kindly like to ask if there could any extension of the page limit to 4/5, as we believe that many reviewers comments could be better addressed while a 2 pages limit restrains.
>
> Thanks in advance

---

### Decision · Program_Chairs · 2023-04-08

Invite to present